# The Impact of Perception–Action Training Devices on Quickness and Reaction Time in Female Volleyball Players

**DOI:** 10.3390/jfmk9030147

**Published:** 2024-08-27

**Authors:** Nicola Mancini, Marilena Di Padova, Rita Polito, Siria Mancini, Anna Dipace, Angelo Basta, Dario Colella, Pierpaolo Limone, Giovanni Messina, Marcellino Monda, Antonietta Monda, Mariasole Antonietta Guerriero, Antonietta Messina, Fiorenzo Moscatelli

**Affiliations:** 1Department of Wellbeing, Nutrition and Sport, Pegaso Telematic University, 80143 Naples, Italy; nicola.mancini.fg@gmail.com (N.M.); fiorenzo.moscatelli@unipegaso.it (F.M.); 2Department of Humanistic Studies, University of Foggia, 71121 Foggia, Italy; marilena.dipadova@unifg.it (M.D.P.); siria_mancini.585707@unifg.it (S.M.); mariasole.guerriero@unifg.it (M.A.G.); 3Department of Clinical and Experimental Medicine, University of Foggia, 71122 Foggia, Italy; rita.polito@unifg.it (R.P.); angelo.basta@unifg.it (A.B.); 4Department of Psychology and Education, Pegaso Telematic University, 80143 Naples, Italy; anna.dipace@unipegaso.it (A.D.); pierpaolo.limone@unipegaso.it (P.L.); 5Department of Biological and Environmental Sciences and Technologies, University of Salento, 73100 Lecce, Italy; dario.colella@unisalento.it; 6Department of Experimental Medicine, Section of Human Physiology and Unit of Dietetics and Sports Medicine, University of Campania “Luigi Vanvitelli”, 80138 Naples, Italy; giovanni.messina@unicampania.it (G.M.); marcellino.monda@unicampania.it (M.M.); 7Department of Human Science and Quality of Life Promotion, San Raffaele Telematic University, 00166 Rome, Italy; antonietta.monda@uniroma5.it; 8Department of Precision Medicine, University of Campania “Luigi Vanvitelli”, 80138 Naples, Italy

**Keywords:** volleyball players, reaction time, light sensors, performances

## Abstract

The objective of this study was to examine the effect of a training program utilizing action perception technology (PAD) tools on improving the motor reaction times and neuromuscular capabilities of the upper and lower limbs compared to a traditional training program. Twenty-four female volleyball players competing in the Italian national championship were randomized into two groups of 12 athletes each: an experimental group (EG) and a control group (CG). A preliminary analysis confirmed the absence of significant differences in age and anthropometric characteristics between the groups. All the players underwent an initial battery of tests (pre-test), including Reaction Time simple Upper and Lower Limb (RTsUL and RTsLL) and Tapping Upper and Lower Limb (TUL and TLL). During a 6-week training program, the experimental group used exercises with a technological system of illuminated disks, while the control group followed the traditional training methods without advanced technology. At the end of the program, both groups were subjected to final tests (post-test). The main results show that after 6 weeks, both groups improved their performance compared to the initial tests. However, EG achieved significantly better results than CG in every test, with significant reductions in average times (ip%) of −14.9% in RTsUL (DX = −0.072 s, t = 23.2, *p* < 0.05, d = 6.7), −14.9% in RTsLL (DX = −0.091 s, t = 44.0, *p* < 0.05, d = 12.7), −10.6% in TUL (DX = −0.622 s, t = 42.0, *p* < 0.05, d = 12.1), and −10.7% in TLL (DX = −0.983 s, t = 43.1, *p* < 0.05, d = 12.4). The use of light-based perception–action technology devices in volleyball training has shown potential for significantly improving movement speed and reaction time. However, further research is needed to determine whether these improvements actually translate into enhanced overall performance in competitive contexts compared to the traditional training methods.

## 1. Introduction

In modern volleyball, the quickness of movement and reaction times are fundamental elements for determining athletes’ success. This sport requires quick responses to visual stimuli and explosive movement capabilities, as the ball can travel at speeds over 100 km/h and change trajectory suddenly. The ability to promptly react to external stimuli and the consequent quickness of movement are crucial during the decisive moments of the game, such as blocking at the net, receiving, defending, and counterattacking, where every millisecond can make a difference.

In situational sports like team sports, it is widely recognized that the ability to react quickly to a stimulus, combined with high levels of coordination, forms a solid foundation for success in competitive situations. Although it is commonly accepted that elite athletes perform better than beginners, it remains unclear whether superior performance results from more advanced sensorimotor coordination [1,2,3].

Quickness is defined as the ability to achieve the highest possible speed of reaction and movement under specific conditions, relying on cognitive processes, maximal efforts, and the functionality of the neuromuscular system [4,5]. It is often a critical aspect for volleyball players, both in attacking and defensive situations. In this study, we considered the following: (a) the speed of individual movements: performing acyclic movements at maximum velocity against minimal resistance; (b) the frequency of movements: performing cyclic movements at maximum velocity against minimal resistance. The acyclic speed of both the upper and lower limbs is crucial in specific technical actions in volleyball, such as arm movement in executing a smash or rapid mobility on the court.

Observing a volleyball match, it is evident how athletes who react quickly and those who are faster in their movements have a competitive advantage, as they can perform complex techniques with greater efficiency and precision. Segmental coordination, essential for sports and daily activities, allows for controlling and synchronizing body movements to perform precise and efficient motor actions [6,7].

During performance, athletes constantly access sensory information to guide and adapt their motor behavior. Recent research has increasingly focused on the role of perceptual–cognitive function (PCF) in both athletes and non-athletes. Visual skills are crucial for developing these perceptual–cognitive abilities [8,9]. Reaction times (RTs) are defined as the interval between the appearance of a stimulus, its perception, and the response [10]. RT is a critical factor in many sports and can be significantly improved with targeted training over time [11,12,13]. Engaging in physical activity and sports enhances RT through a variety of motor actions [14,15,16,17]. In this study, RTs are considered the time from the presentation of a visual stimulus to the response, measured as the quick movement of the hand or foot: RT = Reaction time + Movement time.

The use of technological systems that enhance cognitive skills through visual stimuli is gaining popularity, effectively training and testing reaction times [14,15,18]. Recently, there has been an increased use of technological tools, including wireless devices that emit LED signals or visual cues (such as letters, numbers, or symbols), to train and assess reaction and movement times [18,19,20]. These tools have been found useful both as training aids and for assessment in individual and team sports, studying sensory and cognitive skills as well as motor abilities like reaction speed, quickness, and agility [18,21].

The objective of this study was to examine the performance differences between a training program that utilizes PAD (perception–action technology) tools and a traditional training program (without the use of advanced technologies) on motor reaction times and the quickness of the upper and lower limbs in volleyball players. The main hypothesis is that a training program enriched with specific exercises using technological tools can significantly improve reaction times and the quickness of movement compared to a traditional training program for elite volleyball players. Specifically, it is expected that such technological tools can provide immediate feedback and visual stimuli that facilitate a faster and more measurable improvement in the reaction capabilities and quickness of movement.

Volleyball is a dynamic sport where physical performance, along with technical and tactical factors, significantly impacts success in competitions [22,23,24,25]. Players require moderate to high levels of sensory and cognitive skills, in addition to physical and motor abilities, as fundamental prerequisites. Optimizing physical performance while improving technical and tactical factors of volleyball players is crucial to achieving success in competitions [22,26]. Short reaction times [27], quickness, and movement velocity [14] are essential for success in volleyball. In sports, the ability to react swiftly and move agilely are closely intertwined and continuously interact to execute actions effectively.

The novelty of this study lies in its application of cutting-edge technological tools specifically designed to enhance action perception in the context of volleyball training. While technology has been employed in various sports, this research uniquely focuses on how devices that improve action perception can affect performance in volleyball. By comparing the effects of a traditional training program with one that incorporates these advanced technological tools, the study seeks to provide novel insights into the efficacy of technology-enhanced training methods. This approach has the potential to lead to new training strategies and improve overall performance in volleyball.

## 2. Materials and Methods

### 2.1. Participants

In this study, 24 (effect size = 0.802) female volleyball players who participated in the Italian national championship voluntarily took part. The participants were randomized by random draw into two groups of 12 athletes each: an experimental group (EG) and a control group (CG) (Table 1). Subsequently, preliminary analyses were conducted to compare demographic and anthropometric characteristics between the groups. The results of statistical tests (*t*-test) showed that there were no significant differences between the groups in the baseline variables (*p* > 0.05), indicating that the groups were comparable at the beginning of the study.

The selection criteria were as follows: the participants had to have played competitive volleyball for a minimum of 6 years, completed the training program, and passed all the required tests for the study. The exclusion criteria included the following: any recent injury that required medical treatment, poor health conditions, neurological adverse events such as seizures, incomplete participation in the training and testing program, and having had a COVID-19 infection within the six months prior to the study’s start.

All the procedures performed in this study comply with the directives of the Declaration of Helsinki (2013). Informed consent was required for data collection. The participants were aware that they could withdraw from the study at any time.

### 2.2. Experimental Procedure

The study took place during the pre-season training phase, which involved physical and technical preparation, following a break of about two months after the previous competitive season. During this transitional period, each player was assigned a general program with light to moderate intensity, focusing on maintaining physical fitness and preventing injuries. This allowed for the initiation of the program with significant loads from the early days without the need for an introductory conditioning period.

The initial tests (pre-tests) were administered to all the participants using the Fitlight Trainer technological system [28]; subsequently, each athlete performed a 6-week training program, with the experimental group (EG) using a ReactionX technological system [29] consisting of lighted disks, while the control group (CG) underwent the traditional training without the use of any modern technology. At the end of the 6-week training program, both groups underwent the final tests (post-tests). 

The order of test administration was identical before and after the training intervention and included the reaction time and tapping test. The physical training period included 6 training sessions per week for all the athletes, with 3 days of double sessions (morning and evening) in the first 3 weeks and 4 double sessions in the remaining weeks leading up to the start of the competitive season (Table 2). 

On Mondays, Wednesdays, and Fridays, the CG performed traditional exercises, while the EG carried out exercises using LED light devices. In this study, efforts were made to ensure that there was no overall difference in the training volume between the EG and CG, as it is an important factor when comparing the effects among multiple groups.

### 2.3. ReactionX System

The experimental training program used the ReactionX system (Figure 1), which features up to eight wireless RGB LED lights as targets. The athletes deactivate these lights using various parts of their bodies according to the training routine. The lights are controlled via a tablet with software that allows for programming color, activation, and deactivation through pre-set or custom programs. The lights can be placed at different distances and support points within the training area, such as panels, poles, walls, mirrors, windows, or on the ground. During sessions focused on motor reaction and agility, the participants can deactivate the lights by making contact or passing near them using any part of their body or sports equipment (Figure 1). The individuals can train alone or compete with others, with reaction times and selection errors recorded for immediate feedback. These data can also be saved and exported for further analysis.

### 2.4. Reaction Time Simple Upper Limb (RTsUL) 

The simple upper limb reaction time test (RTsUL; ICC/Rho: 0.94, 95% CI: 0.72–0.98, *p* < 0.001) [30] was used to measure reaction times and the coordination of the upper limbs. The participants stood behind a table with eight LED sensor Fitlight System arranged in a semicircle at a forearm-length distance. The sensors flashed blue randomly, and the participants deactivated them by passing their hands over the light without touching it. Both hands could be used, with no specific instructions on which to use. The hands were placed on the table before each pass. The test lasted 45 s, during which the average response time was calculated. Each participant performed two trials with a 3 min break in between, and the best average time was recorded. This test assessed reaction quickness, attention, visual scanning, and processing speed.

### 2.5. Reaction Time Simple Lower Limb (RTsLL) 

The assessment of basic reaction times and coordination in the lower limbs [30] was carried out using the Reaction Time simple Lower Limb test (RTsLL; ICC/Rho: 0.74 * (95% CI: 0.34 and 0.92), *p* < 0.001) [31].

The subject is positioned in the center of a semicircular arrangement of eight LED lights [17]. The spacing from the feet to the LEDs is equivalent to the length of the lower limb from the ground to the tibial tuberosity (leg). To deactivate the lights, the subject must start from a shoulder-width stance with ground contact. The test duration is 45 s, during which the average response time in milliseconds is recorded for turning off randomly illuminated blue lights by swiftly moving the preferred foot to the corresponding LED. Each participant undertakes two trials with a 3 min rest between them, and the best average time from these trials is considered. The test measures not only quick reactions and attention but also the speed of visual scanning and processing.

### 2.6. Tapping Lower Limb (TLL)

The test is designed to assess how quickly the feet can move (ICC > 0.90) [32]. The subject sits on an adjustable chair with one foot placed on the ground next to a 20 × 5 × 1 cm tablet that is fixed with adhesive tape. The leg should be perpendicular to the ground, with the thigh and leg forming about a 90° angle [33]. When the examiner starts the test, the subject uses their preferred foot to make 40 touches to the right and left sides of the tablet on the ground. While the subject can hold onto the chair for stability, they must not touch the tablet. The Chronojump force platform [34] measures the time in seconds and tenths needed to complete the 40 touches. Each foot is tested twice with a suitable rest period between trials, and the shortest time for each foot is recorded. The final score is the average time, in milliseconds, calculated from the two feet. 

### 2.7. Tapping Upper Limb (TUL)

The test aims to measure the quickness and coordination of movement of the upper limb (ICC 0.88–0.93) [35]. Materials: two 20 cm diameter disks, a table, a 10 × 20 cm plate, measuring tape, stopwatch. Two disks, each with a diameter of 20 cm, are fixed on the surface of a table. The central points of the disks are 80 cm apart (thus, the edges are 60 cm apart). A rectangular plate measuring 10 × 20 cm is placed in the center between the two equidistant disks. The participant stands in front of the table, with the height reaching the hips. The non-dominant hand is positioned on the rectangular plate, while the dominant hand is placed on the opposite disk, crossing the arms [36].

At the command, the athlete moves their dominant hand laterally between the two disks as quickly as possible, passing over the hand placed in the middle each time. The examiner starts the stopwatch and counts out loud 25 touches (touches executed by the dominant hand). The total time to complete the 25 touches for both the dominant and non-dominant hand is recorded. The test is performed twice. The total score is calculated as the average of the milliseconds of timing between the two hands.

### 2.8. Statistical Analysis 

The data were analyzed using IBM SPSS version 25.0 (IBM Inc., Endicott, NY, USA). The distribution of each variable was assessed with Shapiro–Wilk tests. The descriptive statistics included the mean (X), standard deviation (SD), and difference in means (DX). A MANOVA was performed to determine if there were statistically significant differences in at least one of the means between the two groups. Both independent and paired t-tests were used, and Cohen’s d, as well as the percentage increase (ip%), were calculated. Cohen’s d (effect size) is interpreted as follows: 0.1–0.2 indicates a small effect, 0.3–0.5 a medium effect, 0.5–0.8 a large effect, and above 0.8 a very large effect [37]. The threshold for statistical significance was set at *p* < 0.05. The percentage increase (ip%) was calculated using the following formula:ip% = [(Xpost − Xpre)/Xpre] ∗ 100(1)

## 3. Results

Having accepted the hypothesis of the non-equality of the variances, Table 3 reports the results of the RTsUL, RTsLL, TUL, and TLL tests in the pre- and post intervention. 

In the pre-test phase, no significant differences were found between the average times (DX) obtained in all the tests (RTsUL (s), DX = −0.002 s, *p* = 0.562; RTsLL (s), DX = −0.007 s, *p* = 0.582; TUL (s), DX = 0.059 s, *p* = 0.580; TLL (s), DX = 0.05 s, *p* = 0.795). 

The data shows that before the start of the training program, the groups achieved similar average times in each trial, so both groups started from the same level of performance.

On the other hand, the results obtained in the post-test phase (Table 3) show significant differences in the mean times (DX) in all the tests (RTsUL (s), DX = −0.062 s, *p* < 0.05; RTsLL (s), DX = −0.07 s, *p* < 0.05; TUL (s), DX = −0.432 s, *p* < 0.05; TLL (s), DX = −0.84 s, *p* < 0.05). Since *p* < 0.05 in all the tests, we can reject the null hypothesis of mean homogeneity and therefore conclude that the observed differences in both groups are not due to chance but to the effect of training. 

Observing the mean differences between pre- and post-tests (Figure 2 and Figure 3) in each group, we can see a significant improvement in the mean execution times in all the tests, even if with different magnitudes between EG and CG.

In Table 4 we have reported the performance improvements of EG with significant decreases in mean times (ip%) of −14.9% in the RTsUL (DX = −0.072 s, t = 23.2, *p* < 0.05, d = 6.7), −14.9% in the RTsLL (DX = −0.091 s, t = 44, *p* < 0.05, d = 12.7), −10.6% in the TUL (DX = −0.622 s, t = 42.0, *p* < 0.05, d = 12.1), and −10.7% in the TLL (DX = −0.983 s, t = 43.1, *p* < 0.05, d = 12.4). 

The value of Cohen‘s d highlights that the relationship between the variables in the EG is very strong in all the tests as its value is much higher than 0.8 [38]. Within the limit of the sample analyzed, it can, therefore, be argued that in the present study, the decrease in average times in all the tests is associated with the effects of the training program with the use of exercises that make use of perception–action devices.

The CG achieves significant but minor performance improvements (Table 4), with decreases in mean times (ip%) of −2.5% in the RTsUL (DX = −0.012 s, t = 9.6, *p* < 0.05, d = 2.8), −4.5% in the RTsLL (DX = −0.028 s, t = 11.0, *p* < 0.05, d = 3.2), −2.3% in the TUL (DX = −0.131 s, t = 6.2, *p* <0.05, d = 1.8), and −1.0% in the TLL (DX = −0.093 s, t = 2.8, *p* < 0.05, d = 0.8). 

Although the improvements in mean CG execution times recorded in the tests are small, it should be noted that the calculated Cohen’s d value is classified as very large (d > 0.8) (Figure 4). Therefore, we can argue that in the present study, a traditional training program can be associated with an improvement in performance, even if slight, in the trials administered.

## 4. Discussion

The primary outcomes of our study reveal that following 6 weeks of training, both the experimental group (EG) and the control group (CG) showed improvements in their performances across all the tests compared to their initial baseline measurements. However, in the post-test assessments, the EG consistently demonstrated significantly better results than the CG in each test (Figure 2 and Figure 3). These findings underscore the effectiveness of the training program, particularly within the EG.

Hodges et al. suggest that the speed of movement does not only depend on the physical abilities of individuals, but also on cognitive factors that play a crucial role [39].

In our study, the positive effects of training were evident in the tests assessing reaction time and eye–foot movement quickness within the EG. Additionally, tapping tests, a widely used method for evaluating neuromuscular system capabilities during small-amplitude movements, revealed positive outcomes by reducing the influence of muscle contraction speed [40].

From a methodological standpoint, our results confirm that a 6-week training program incorporating PAD sensors is sufficient to enhance specific Performance-Related Cognitive Functions (PCFs). 

Comparable studies conducted by other authors, such as Ciesluk et al. [41], Schwab and Memmert [42], and Wimshurst et al. [43], have yielded similar results with training programs lasting between 6 and 8 weeks.

The RTsUL (Reaction Time Upper Limb) and RTsLL (Reaction Time Lower Limb) tests for the EG exhibited a substantial improvement margin of 15%. This is particularly noteworthy as a low reaction time is considered the initial crucial step for effectively performing tasks requiring agility and quickness [44].

Our study’s findings align with those of Florkiewicz et al. [45], where 28 university students in an experimental group (male, n = 10, female, n = 5) and a control group (male, n = 8, female, n = 5) significantly improved their eye–hand reaction–execution time (*p* < 0.001) after 6 weeks of training using the Fitlight trainer system. Furthermore, Bidil et al. [46] demonstrated that 8 weeks of training with perception–action light sensors resulted in improved reaction times among international badminton athletes. The results of our study highlight the importance of transversal training that stimulates not only the conditional component, but also the cognitive and attentional one. The ability to switch between tasks requiring different levels of intellect and shift one’s center of focus is known as alternate attention. One essential component of selective attention is the capacity to shift focus from one region to another in response to the many environmental conditions that exist. The process of focusing on a specific object in the environment for a set amount of time is known as selective attention, and it is influenced by several environmental conditions. Selective attention enables us to ignore unnecessary things and concentrate on what really matters because attention is a finite resource. Training creates long-lasting imprinted habits in the adult brain system that especially enable the accurate performance of difficult motor tasks [3]. Athletes must be able to identify the visual fields that hold the greatest amount of information, concentrate their attention where it is most needed, and deduce meaning from these fields quickly and accurately. Successful sporting performance may depend on knowing where to look and when to look, especially in broad scenes where information relevant to the task can be found in multiple locations. The ability to remember and identify a play pattern that is developing is the most trustworthy measure of anticipatory competence in team ball sports. For example, it is impossible to compare the anticipation of a setting action, where the goal is to position the ball in the best possible position for the attack, with that of a volleyball serve, when the primary goal is to send the ball over the net and into the opposing court [47]. Because of this, players could have to make an anticipatory decision based on perceived event likelihood in some situations, while in others they might have to rely entirely on their ability to decipher information from an opponent’s postural orientation. The type of action being analyzed may affect one’s ability to extract meaningful information from an athletic event, even within the same sport. 

Our study had certain limitations because we only looked at younger female volleyball players as study subjects. In the future, it could be interesting to look into male volleyball players as well in order to rule out the influence of hormonal shifts. In addition, a study involving male athletes and non-athletes ought to be carried out. Furthermore, the data in this study has not been compared to that of the general population; therefore, this is an area that needs to be explored in the future to determine whether the findings apply only to volleyball players, or even only to female volleyball players, or to the overall population. Lastly, it is important to keep in mind that the volleyball players’ exams had little to do with the duties involved in volleyball, thus the findings might not apply to other situations.

### Practical Implications and Further Research

Exercises enriched with PAD, when adapted to the specific performance model of the sport and employing appropriate methodologies, can produce superior results in certain contexts compared to conventional training approaches.

This aspect is particularly pertinent for those involved in physical preparation, where there is often a critical need to rapidly enhance performance levels during the pre-season phase. Therefore, coaches and quickness and conditioning specialists should consider integrating perception–action light sensors into their training methodologies to optimize sports performance. Nonetheless, further research is warranted, particularly to validate the long-term effects with larger athlete samples encompassing various sports disciplines and both genders. Additionally, controlled environment testing, such as laboratory-based assessments of physiological parameters, should be incorporated to aid in developing standardized testing protocols that incorporate perception–action technologies.

## 5. Conclusions

The use of light-based perception–action technology devices, integrated into a 6-week training program, has led to significant improvements in both movement speed and motor reaction time tests among volleyball players. These advanced technological tools provided immediate and precise feedback, stimulating the athletes’ visual and cognitive systems, and allowing them to react more quickly and efficiently to external stimuli. By refining these critical performance aspects, it is expected that athletes will be able to execute volleyball techniques with greater speed and accuracy, gaining a competitive advantage on the field. However, further research is needed to determine whether these improvements translate into overall enhanced performance in competitive contexts compared to traditional training methods.

## Figures and Tables

**Figure 1 jfmk-09-00147-f001:**
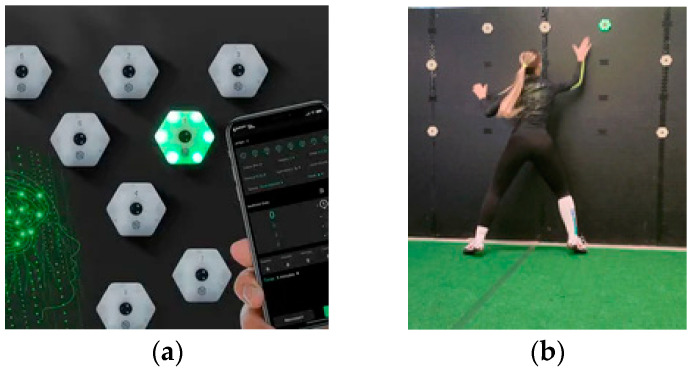
ReactionX system features up to eight wireless RGB LED lights as targets (**a**); and athlete in training (**b**).

**Figure 2 jfmk-09-00147-f002:**
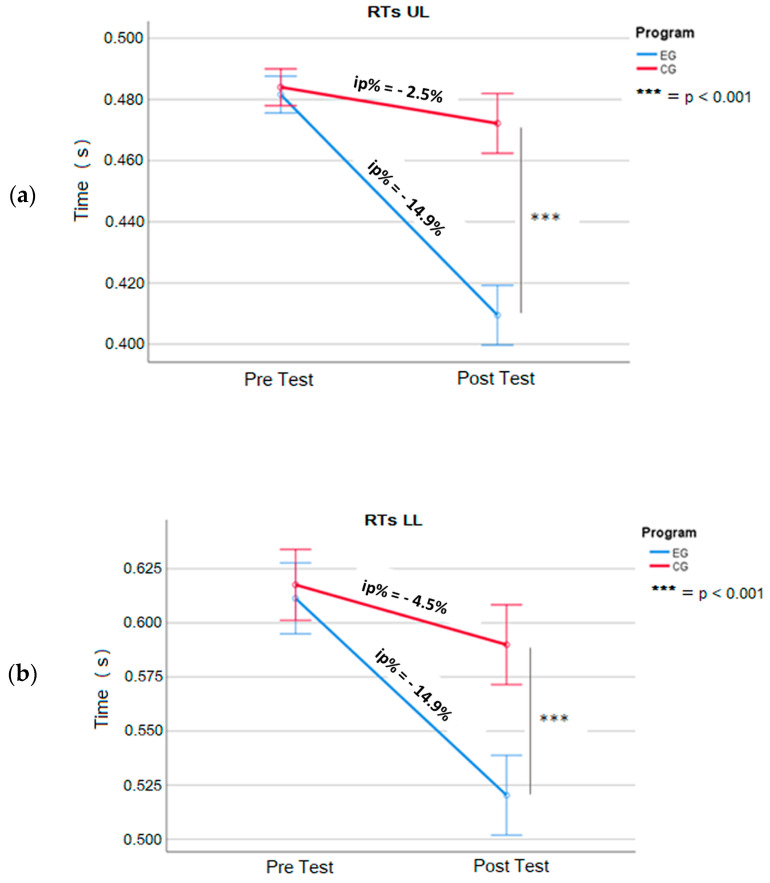
Mean scores for Reaction Time simple Upper Limb (RTsUL) (**a**) and Reaction Time simple Lower Limb (RTsLL) (**b**). Experimental group vs. control croup (EG vs. CG), pre- and post training. Error bar: 95% CI. ip%= increase percentage; (s) = second.

**Figure 3 jfmk-09-00147-f003:**
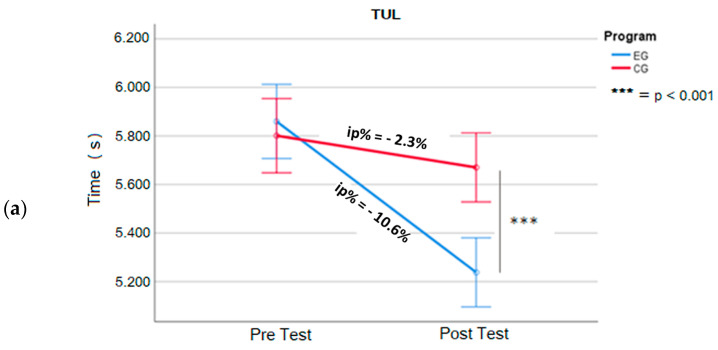
Mean scores for Tapping Upper Limb (TUL) (**a**) and Tapping Lower Limb (TLL) (**b**). Experimental vs. control group (EG vs. CG), pre- and post training. Error bar: 95% CI. ip%= increase percentage; (s) = second.

**Figure 4 jfmk-09-00147-f004:**
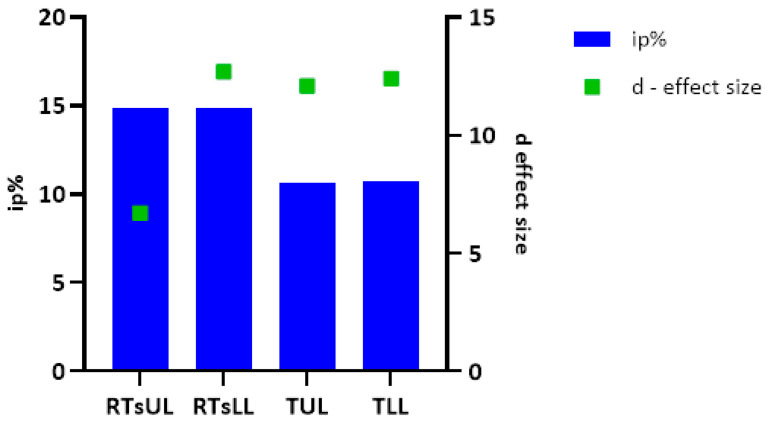
Values of percentage increase (ip%) and side effect d—Cohen’s (d) in the tests RTsUL, RTsLL, TUL, and TLL in EG. RTsUL = Reaction Time simple Upper Limb; RTsLL = Reaction Time simple Lower Limb; TUL = Tapping Upper Limb; TLL = Tapping Lower Limb; EG = experimental group.

**Table 1 jfmk-09-00147-t001:** Anthropometric characteristics.

Parameters	Experimental Group (n = 12)	Control Group (n = 12)	*p*-Value
Age	20.3 ± 1.1	20.4 ± 1	0.686
Height (cm)	175.2 ± 5.1	176.1 ± 4.9	0.657
Body mass (kg)	70.0 ± 7.3	68.7 ± 5.8	0.647

Values are expressed as mean ± standard deviation. cm = centimeters; kg = kilograms.

**Table 2 jfmk-09-00147-t002:** Overview of the overall weekly training schedule.

	Monday	Tuesday	Wednesday	Thursday	Friday	Saturday
Morning session from 9.30 to 11.00 a.m.	Physical training Motor reaction, agility, and quickness.	Physical training Strength training	Physical training Motor reaction, agility, and quickness.	Physical training Strength training	Physical training Motor reaction, agility, and quickness.	Strength training and friendly match on the sand
Afternoon session from 7.00 to 9.00 p.m.	Technical and tactical training	Rest	Technical and tactical training	Rest	Technical and tactical training	Rest

**Table 3 jfmk-09-00147-t003:** *t*-test results for independent groups EG and CG in RTsUL, RTsLL, TUL, and TLL in pre- and post intervention (n = 24).

Test	Group	X ± SD	DX	95% CI Lower Upper	*t*	*p*	*d*
RTsUL (s) pre	EG (n = 12)	0.482 ± 0.008	−0.002	−0.011 0.006	−0.59	0.562	0.18
	CG (n = 12)	0.484 ± 0.011					
RTsUL (s) post	EG (n = 12)	0.410 ± 0.019	−0.062	−0.077 −0.049	−9.4	0.000 *	4.77
	CG (n = 12)	0.472 ± 0.013					
RTsLL(s) pre	EG (n = 12)	0.611 ± 0.027	−0.007	−0.029 0.017	−0.56	0.582	0.26
	CG (n = 12)	0.618 ± 0.027					
RTsLL (s) post	EG (n = 12)	0.520 ± 0.032	−0.07	−0.096 −0.043	−5.53	0.000 *	2.41
	CG (n = 12)	0.590 ± 0.029					
TUL (s) pre	EG (n = 12)	5.860 ± 0.244	0.059	−0.158 0.275	0.56	0.580	0.22
	CG (n = 12)	5.801 ± 0.266					
TUL (s) post	EG (n = 12)	5.238 ± 0.232	−0.432	−0.633 −0.231	−4.46	0.000 *	1.78
	CG (n = 12)	5.670 ± 0.243					
TLL (s) pre	EG (n = 12)	9.153 ± 0.489	0.05	−0.345 0.445	0.26	0.795	0.11
	CG (n = 12)	9.103 ± 0.444					
TLL (s) post	EG (n = 12)	8.170 ± 0.473	−0.840	−1.202 −0.478	−4.81	0.000 *	2.23
	CG (n = 12)	9.010 ± 0.376					

RTsUL—Reaction Time simple Upper Limb; RTsLL—Reaction Time simple Lower Limb; TUL—Tapping Upper Limb; TLL—Tapping Lower Limb; EG—experiment group; CG—control group; X ± DS—mean ± standard deviation; DX—difference in means; 95% C.I.—interval of confidence with lower and upper levels; t—Student’s *t*-test; *p*—statistical level of probability; d—Cohen’s effect size. * Significant at *p* < 0.05; T—value of t at the significance level of 0.05 = 2.201.

**Table 4 jfmk-09-00147-t004:** *t*-test results for paired samples EG and CG in RTsUL, RTsLL, TUL, and TLL in pre- and post intervention.

Test	Group	Phase of Test	X ± SD	DX Post-Pre ± SD	95% CI Lower Upper	t	*p*	d	ip%
RTsUL (s)	EG (n = 12)	pre	0.482 ± 0.008	0.072 ± 0.011	0.065 0.079	23.2	0.000 *	6.7	−14.9
		post	0.410 ± 0.019						
	CG (n = 12)	pre	0.484 ± 0.011	−0.012 ± 0.004	0.009 0.015	9.6	0.000 *	2.8	−2.5
		post	0.472 ± 0.013						
RTsLL (s)	EG (n = 12)	pre	0.611 ± 0.027	−0.091 ± 0.007	0.086 0.096	44.0	0.000 *	12.7	−14.9
		post	0.520 ± 0.032						
	CG (n = 12)	pre	0.618 ± 0.027	−0.028 ± 0.009	0.022 0.033	11.0	0.000 *	3.2	−4.5
		post	0.590 ± 0.029						
	EG (n = 12)	pre	5.860 ± 0.244	−0.622 ± 0.051	0.589 0.654	42.0	0.000 *	12.1	−10.6
TUL (s)		post	5.238 ± 0.232						
	CG (n = 12)	pre	5.801 ± 0.266	−0.131 ± 0.073	0.085 0.177	6.2	0.000 *	1.8	−2.3
		post	5.670 ± 0.243						
	EG (n = 12)	pre	9.153 ± 0.489	−0.983 ± 0.079	0.932 1.033	43.1	0.000 *	12.4	−10.7
TLL (s)		post	8.170 ± 0.473						
	CG (n = 12)	pre	9.103 ± 0.444	−0.093 ± 0.115	0.019 0.166	2.8	0.018 *	0.8	−1.0
		post	9.010 ± 0.376						

RTsUL—Reaction Time simple Upper Limb; RTsLL—Reaction Time simple Lower Limb; TUL—Tapping Upper Limb; TLL—Tapping Lower Limb; EG—experiment group; CG—control group; X ± DS—mean ± standard deviation; DX—difference in means; 95% C.I.—interval of confidence with lower and upper levels; t—Student’s *t*-test; *p*—statistical level of probability; d—Cohen‘s effect size; ip%— increase percentage; * Significant at *p* < 0.05; *T*—value of t at the significance level of 0.05 = 2.074.

## Data Availability

The original contributions presented in the study are included in the article, further inquiries can be directed to the corresponding author/s.

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
