# Peer review of "The Impact of Perception–Action Training Devices on Quickness and Reaction Time in Female Volleyball Players"

_jfmk, 2024, doi:10.3390/jfmk9030147_

Round 1

Reviewer 1 Report

Comments and Suggestions for Authors

Evaluation of manuscript jfmk-3146102

This manuscript evaluated the effect of 6-wk of PAD training on volleyball performance. I will present bellow my considerations about the text in view to help the authors to improve the quality of manuscript.

The title is generic, I suggest improve the title aiming to present the main result of the study.

The abstract is inadequate, please review and insert the numeric and statistical results.

The introduction is completely disorganized, it talks about the sport, then agility, then volleyball. It is important to organize the content. Furthermore, we do not have an introduction to the hypothesis and the original aspects. In my understanding, this use of technology is not recent in sport. The authors should highlight the original aspects to justify the publication of this study.

How was the sample representativeness calculation performed? This data must be included in the study.

Authors must make these adjustments before resubmitting the manuscript for evaluation.

Comments on the Quality of English Language

Adequate english language.

Author Response

Reviewer 1

This manuscript evaluated the effect of 6-wk of PAD training on volleyball performance. I will present bellow my considerations about the text in view to help the authors to improve the quality of manuscript.

Reply

Dear reviewer, thank you for your comment, your suggestions will greatly improve our manuscript.

Reviewer 1

The title is generic, I suggest improve the title aiming to present the main result of the study.

Reply

Dear reviewer, thank you for your suggestion. We have changed the title as follows:

The impact of perception-action training devices on quickness and reaction time in female volleyball players

Reviewer 1

The abstract is inadequate, please review and insert the numeric and statistical results.

Reply

Dear reviewer, thank you for your suggestion. We have modified the abtsract as follows:

The objective of this study was to examine the effect of a training program utilizing action perception technology (PAD) tools on improving motor reaction times and neuromuscular capabilities of the upper and lower limbs, compared to a traditional training program. Twenty-four female volleyball players competing in the Italian national championship were randomized into two groups of 12 athletes each: an experimental group (EG) and a control group (CG). A preliminary analysis confirmed the absence of significant differences in age and anthropometric characteristics between the groups. All players underwent an initial battery of tests (pre-test), including Reaction Time simple Upper and Lower Limb (RTsUL and RTsLL) and Tapping Upper and Lower Limb (TUL and TLL). During a 6-week training program, the experimental group used exercises with a technological system of illuminated discs, while the control group followed traditional training methods without advanced technology. At the end of the program, both groups were subjected to final tests (post-test). The main results show that after 6 weeks, both groups improved their performance compared to the initial tests. However, EG achieved significantly better results than CG in every test, with significant reductions in average times (ip%) of -14.9% in RTsUL (DX = -0.072 s, t = 23.2, p < 0.05, d = 6.7), -14.9% in RTsLL (DX = -0.091 s, t = 44.0, p < 0.05, d = 12.7), -10.6% in TUL (DX = -0.622 s, t = 42.0, p < 0.05, d = 12.1), and -10.7% in TLL (DX = -0.983 s, t = 43.1, p < 0.05, d = 12.4). In sommary, the use of light-based perception-action technology devices, integrated into a 6-week training program, led to significant improvements in both movement quickness and motor reaction speed among female volleyball players, demonstrating greater effectiveness compared to traditional training methods.

Reviewer 1

The introduction is completely disorganized, it talks about the sport, then agility, then volleyball. It is important to organize the content. Furthermore, we do not have an introduction to the hypothesis and the original aspects. In my understanding, this use of technology is not recent in sport. The authors should highlight the original aspects to justify the publication of this study.

Reply

Dear reviewer, thank you for your suggestion. We have modified the introduction as follows:

In modern volleyball, quickness of movement and reaction times are fundamental elements for determining athletes' success. This sport requires quick responses to visual stimuli and explosive movement capabilities, as the ball can travel at speeds over 100 km/h and change trajectory suddenly. The ability to promptly react to external stimuli and the consequent quickness of movement are crucial during decisive moments of the game, such as blocking at the net, receiving, defending, and counterattacking, where every millisecond can make a difference.

In situational sports like team sports, it is widely recognized that the ability to react quickly to a stimulus, combined with high levels of coordination, forms a solid foundation for success in competitive situations. Although it is commonly accepted that elite athletes perform better than beginners, it remains unclear whether superior performance results from more advanced sensorimotor coordination [1].

Quickness is defined as the ability to achieve the highest possible speed of reaction and movement under specific conditions, relying on cognitive processes, maximal efforts, and the functionality of the neuromuscular system [15]. It is often a critical aspect for volleyball players, both in attacking and defensive situations. In this study, we considered: a) the speed of individual movements: performing acyclic movements at maximum velocity against minimal resistance; b) the frequency of movements: performing cyclic movements at maximum velocity against minimal resistance.The acyclic speed of both upper and lower limbs is crucial in specific technical actions in volleyball, such as the arm movement in executing a smash or rapid mobility on the court.

Observing a volleyball match, it is evident how athletes who react quickly and those who are faster in their movements have a competitive advantage, as they can perform complex techniques with greater efficiency and precision. Segmental coordination, essential for sports and daily activities, allows for controlling and synchronizing body movements to perform precise and efficient motor actions [16].

During performance, athletes constantly access sensory information to guide and adapt their motor behavior. Recent research has increasingly focused on the role of perceptual-cognitive function (PCF) in both elite and non-elite sports. Visual skills are crucial for developing these perceptual-cognitive abilities [2]. Reaction times (RTs) are defined as the interval between the appearance of a stimulus, its perception, and the response [3]. RT is a critical factor in many sports and can be significantly improved with targeted training over time [4]. Engaging in physical activity and sports enhances RT through a variety of motor actions [5–8]. In this study, RTs are considered the time from the presentation of a visual stimulus to the response, measured as the quick movement of the hand or foot: RT = Reaction time + Movement time.

The use of technological systems that enhance cognitive skills through visual stimuli is gaining popularity, effectively training and testing reaction times [2]. Recently, there has been an increased use of technological tools, including wireless devices that emit LED signals or visual cues (such as letters, numbers, or symbols), to train and assess reaction and movement times [9]. These tools have been found useful both as training aids and for assessment in individual and team sports, studying sensory and cognitive skills as well as motor abilities like reaction speed and agility [2, 10].

This study aims to explore how employing advanced technologies, such as action perception devices (PAD), can enhance training effectiveness and positively impact athletic performance in volleyball. The main hypothesis is that a training program enriched with specific exercises using technological tools can significantly improve reaction times and quickness of movement compared to a traditional training program for elite volleyball players. Specifically, it is expected that such technological tools can provide immediate feedback and visual stimuli that facilitate a faster and measurable improvement in reaction capabilities and quickness of movement.

Volleyball is a dynamic sport where physical performance, along with technical and tactical factors, significantly impacts success in competitions [11]. Players require moderate to high levels of sensory and cognitive skills, in addition to physical and motor abilities, as fundamental prerequisites. Optimizing physical performance while improving technical and tactical factors of volleyball players is crucial to achieve success in competitions [12]. Short reaction times [13], speed, and movement velocity [14] are essential for success in volleyball. In sports, the ability to react swiftly and move agilely are closely intertwined and continuously interact to execute actions effectively.

Original Aspects: The originality of this study lies in its application of cutting-edge technological tools specifically designed to enhance action perception in the context of volleyball training. While technology has been employed in various sports, this research uniquely focuses on how devices that improve action perception can affect performance in volleyball. By comparing the effects of a traditional training program with one that incorporates these advanced technological tools, the study seeks to provide novel insights into the efficacy of technology-enhanced training methods. This approach has the potential to lead to new training strategies and improve overall performance in volleyball.

Reviewer 1

How was the sample representativeness calculation performed? This data must be included in the study.

Reply

Dear reviewer, thank you for your comment.

To ensure the representativeness of the sample in the study, the following strategies were employed:

  1. Participant Selection: The sample consisted exclusively of female volleyball players participating in the national championship who met specific inclusion criteria, including at least 6 years of competitive experience, completion of the training program, and passing all required tests. This ensured that participants had similar levels of experience and skill.
  2. Demographic Control: A preliminary analysis was conducted to confirm that there were no significant differences in age and anthropometric characteristics between the control group (CG) and the experimental group (EG). This ensured that the groups were comparable and that any observed effects could be attributed to the experimental conditions rather than baseline differences.

Table 1. Anthropometric characteristics.

Parametrers

Experimental Group (n=12)

Control

Group (n=12)

p-value

Age

20.3 ± 1.1

20.4 ±1

0.686

Height (cm)

175.2 ± 5.1

176.1 ± 4.9

0.657

Body mass (kg)

70.0 ± 7.3

68.7 ± 5.8

0.647

Values are expressed as mean ± standard deviation.

  1. Sample Size Calculation: The sample size of 24 players (12 per group) was determined based on the minimum number needed to achieve adequate statistical power and reliable results. Previous studies and literature reviews guided the preliminary estimation of the required sample size to detect significant differences between groups, aiming to minimize the risk of Type I and Type II errors.
  2. Exclusion Criteria: Participants were excluded if they had recent injuries requiring medical attention, compromised health, neurological adverse events (e.g., seizures), incomplete participation in the training and testing program, or had a COVID-19 infection in the six months prior to the study. These criteria helped maintain sample homogeneity and representativeness, ensuring participants were in optimal condition for analysis.
  3. Statistical Analysis: Statistical tests were used to evaluate the representativeness and comparability of the groups by analyzing demographic and anthropometric characteristics, ensuring that the groups did not differ significantly on key variables. The results of statistical tests (t-test) showed that there were no significant differences between the groups in the baseline variables (p > 0.05), indicating that the groups were comparable at the beginning of the study." This confirmed that observed differences in results could be attributed to the experimental interventions rather than confounding factors.

These measures ensured that the sample was representative and that the study results could be generalized to the population of female volleyball players with similar characteristics.

Reviewer 2 Report

Comments and Suggestions for Authors

Notes to the authors

I sent my observations and suggestions in the attachment.

Author Response

Reviewer 2

 L25-28:

The aim of this study was to investigate the effects of a training method enriched with specific exercises utilizing action perception technological tools (PAD) on improving reaction times, neuromuscular capabilities of the lower and upper limbs, and quickness in a sample of young volleyball players, compared to a training program using traditional methodology. ???

L66-68

This study aims to demonstrate how employing methodologies that utilize cutting-66 edge technological tools can create innovative solutions to enhance training effectiveness, 67 thereby positively impacting athletic performance. ???

Abstract:

Cognitive performance???? What was this cognitive performance measured against? See the wording "enhanced both physical and, to some extent, cognitive performance"!!!

Reply

Dear reviewer, thank you for your comment, your suggestions will greatly improve our manuscript.

We have revised the section to more precisely specify our main objective:

The objective of this study was to examine the effect of a training program utilizing action perception technology (PAD) tools on improving motor reaction times and neuromuscular capabilities of the upper and lower limbs, compared to a traditional training program.

To clarify, the term "cognitive performance" in the context of our study refers to the brain's ability to perceive, process information, and respond quickly to visual stimuli generated by LED action perception devices. These devices have the capability to measure the time (ms) from the appearance of the visual stimulus to the moment of contact to turn off the light. We have considered reaction time in a simpler form compared to the physiological conception, which would have required more sophisticated laboratory tools such as a signal generator and an electromyograph. In this study, RTs are considered as the time from the presentation of a visual stimulus to the response, measured as the quick movement of the hand or foot: RTs = Reaction time + Movement time.

In other words, motor reaction time (RT) can be defined as the time that elapses from when a stimulus appears until a response is initiated, and it is considered a good measure to assess the cognitive system's ability to process information [1,2].

Jensen, A. R. (2006). Clocking the mind: Mental chronometry and individual differences. Elsevier.

Kuang, S. (2017). Is reaction time an index of white matter connectivity during training? Cognitive Neuroscience, 8(2), 126–128.

We hope this explanation clarifies our approach and the measurement of cognitive performance in our study. If you have any further questions or need additional details, We would be happy to provide more information.

Reviewer 2

  1. Introduction

- I do not agree with this aspect "... it remains unclear whether higher performance is the result of more advanced sensorimotor coordination [1]". If you want to support such a point of view, please argue with many more references, not with a single study and that of the authors of this article. If you want to maintain this aspect, then adapt it to specific athletes, do not generalize to all ages and all sports disciplines.

- "During performance, athletes consistently access sensory information to guide and adapt their motor behavior" - reference should be made to this idea.

Reply

Dear reviewer, thank you for your comment. Following your suggestion we have added two bibliographical references to support what has been written. Below I report the text with the new references:

In situational sports like team sports, it is widely recognized that the ability to react quickly to a stimulus, combined with high levels of coordination, forms a solid foundation for success in competitive situations. Although it is commonly accepted that elite athletes perform better than beginners, it remains unclear whether superior performance results from more advanced sensorimotor coordination [1–3].

Reviewer 2

 L50: "elite and non-elite sports disciplines" ??? details please What would be the "non-elite" sports disciplines?

Reply

Dear reviewer, thank you for your comment. These studies were done with athletes and non-athletes, so what is written is a typo. Below is the revised paragraph.

During performance, athletes constantly access sensory information to guide and adapt their motor behavior. Recent research has increasingly focused on the role of perceptual-cognitive function (PCF) in both athletes and non-athletes.

Reviewer 2

L55: "RT is a critical factor in many sports, and with targeted training over time, it can be significantly improved" - what would significant mean, from the perspective of the authors of this manuscript?. Because reaction time (motor reaction latency) is dependent on genetic factors, the nervous system and the alternation of excitation with inhibition, factors that cannot be trained. I can't agree with that statement, especially since it's backed up by a single, 12-year-old study.

Reply

Dear reviewer, thank you for your comment, your suggestions will greatly improve our manuscript.

Always based on the fact that in this study, RTs are considered as the time from the presentation of a visual stimulus to the response, measured as the rapid movement of the hand or foot: RTs = Reaction time + Movement time.

We based ourselves on the contents of the following studies that we report below:

RT can be trained  (Wilke et al.,2020, Balkó  et al., 2017, Yildirim et al., 2020, Turna, 2020), and physical activity and sports allow for the development of a wide variety of actions that can influence its improvement (Gierczuk et al., 2017, Lynall et al., 2018; Walton et al., 2018).

It has been highlighted that physical activity and sports can be related to improved RT (Jain et al., 2015; Okubo et al., 2017; van de Water et al., 2017; Walton et al., 2018)

 References

  1. Wilke, J.; Vogel, O. Computerized Cognitive Training with Minimal Motor Component Improves Lower Limb  Choice-Reaction Time. J. Sports Sci. Med. 2020, 19, 529–534.
  2. Balkó, Š.; Rous, M.; Balkó, I.; Hnízdil, J.; Borysiuk, Z. Influence of a 9-Week Training Intervention on the Reaction Time of Fencers Aged 15 to 18 Years. Phys. Act. Rev. 2017, 5, 146–154. DOI: 10.16926/par.2017.05.19
  3. Yildirim, Y.; Kizilet, A. The Effect of Different Learning Method on the Visual Reaction Time of Hand and Leg in High School Level Tennis Trainees. J. Educ. Issues 2020, 6, 414–424. DOI: 10.5296/jei.v6i2.17970
  4. Turna, B. The Effect of Agility Training on Reaction Time in Fencers. J. Educ. Learn. 2020, 9, 127–135. DOI: 10.5539/jel.v9n1p127

Gierczuk, D.; Lyakh, V.; Sadowski, J.; Bujak, Z. Speed of Reaction and Fighting Effectiveness in Elite Greco-Roman Wrestlers. Percept. Mot. Skills 2017, 124, 200–213.

Lynall, R. C., Blackburn, J. T., Guskiewicz, K. M., Marshall, S. W., Plummer, P., & Mihalik, J. P. (2018). Reaction time and joint kinematics during functional movement in recently concussed individuals. Archives of Physical Medicine and Rehabilitation, 99(5), 880–886.

Jain, A., Bansal, R., Kumar, A., & Singh, K. D. (2015). A comparative study of visual and auditory reaction times on the basis of gender and physical activity levels of medical first year students. International Journal of Applied and Basic Medical Research, 5(2), 124.

Okubo, Y., Schoene, D., & Lord, S. R. (2017). Step training improves reaction time, gait and balance and reduces falls in older people: a systematic review and meta-analysis. British Journal of Sports Medicine, 51(7), 586–593.

van de Water, T., Huijgen, B., Faber, I., & Elferink-Gemser, M. (2017). Assessing cognitive performance in badminton players: a reproducibility and validity study. Journal of Human Kinetics, 55(1), 149–159

Walton, C. C., Keegan, R. J., Martin, M., & Hallock, H. (2018). The potential role for cognitive training in sport: more research needed. Frontiers in Psychology, 9, 1121

We hope this explanation clarifies our approach and the measurement of cognitive performance in our study. If you have any further questions or need additional details, I would be happy to provide more information.

Reviewer 2

L66-70 does not correspond to L25-28. The purpose of the study is different, unclear if both formulations remain. I recommend clarification.

Reply

Dear reviewer, these are two consequential effects that we have clarified better by specifying the main hypothesis

This study aims to explore how employing advanced technologies, such as action perception devices (PAD), can enhance training effectiveness and positively impact athletic performance in volleyball. The main hypothesis is that a training program enriched with specific exercises using technological tools can significantly improve reaction times and speed of movement compared to a traditional training program for elite volleyball players. Specifically, it is expected that such technological tools can provide immediate feedback and visual stimuli that facilitate a faster and measurable improvement in reaction capabilities and speed of movement.

Reviewer 2

 L70-71: "These tools were designed to improve reaction times and agility in a group of young volleyball players, compared to a traditional training program" - I think the authors should reformulate, because these devices were not specially designed for volleyball.

Reply

Dear reviewer, we made the following change

Specifically, it is expected that such technological tools can provide immediate feedback and visual stimuli that facilitate a faster and measurable improvement in reaction capabilities and quickness of movement.

Reviewer 2

 L75-77: "Numerous scientific studies aim to optimize volleyball players' physical performance, crucial alongside technical-tactical factors for achieving 76 success in

competitions [12]." - what are these numerous studies, because the authors only refer to one study and that from 17 years ago???

Reply

Dear reviewer, the period was badly pre-phased. Thank you for highlighting it.

Optimizing physical performance while improving technical and tactical factors of volleyball players is crucial to achieve success in competitions [12].

Reviewer 2

L87: "Schiffer's (1993)" - the reference is not passed at the end. I recommend clarification and inclusion in the "references" section for the source that is referred to here

Reply

Dear Reviewer, thank you for pointing that out. It was a typographical error, and we have corrected it as follows:

In this study, we considered: a) the speed of individual movements: performing acyclic movements at maximum velocity against minimal resistance; b) the frequency of movements: performing cyclic movements at maximum velocity against minimal resistance.

Reviewer 2

  1. Materials and Methods - The simple size?? For the relevance of the number of subjects?

Reply

Dear reviewer, thank you for your suggestion.

We have changed the following:

In this study, 24 female volleyball players who participated in the Italian national championship voluntarily took part. The participants were randomized by random draw into two groups of 12 athletes each: an experimental group (EG) and a control group (CG) (Table 1). Subsequently, preliminary analyses were conducted to compare demographic and anthropometric characteristics between the groups. The results of statistical tests (t-test) showed that there were no significant differences between the groups in the baseline variables (p > 0.05), indicating that the groups were comparable at the beginning of the study.

                                Table 1. Anthropometric characteristics.

Parametrers

Experimental Group (n=12)

Control

Group (n=12)

p-value

Age

20.3 ± 1.1

20.4 ±1

0.686

Height (cm)

175.2 ± 5.1

176.1 ± 4.9

0.657

Body mass (kg)

70.0 ± 7.3

68.7 ± 5.8

0.647

Values are expressed as mean ± standard deviation.

The selection criteria were as follows: participants had to have played competitive volleyball for a minimum of 6 years, completed the training program, and passed all required tests for the study. Exclusion criteria included: any recent injury that required medical treatment, poor health conditions, neurological adverse events such as seizures, incomplete participation in the training and testing program, and having had a COVID-19 infection within the six months prior to the study's start.

Our choice was based on a series of considerations that we illustrate in more detail. To ensure the representativeness of the sample in the study, the following strategies were used:

Participant Selection: The sample consisted exclusively of female volleyball players participating in the national championship who met specific inclusion criteria, including at least 6 years of competitive experience, completion of the training program, and passing all required tests. This ensured that participants had similar levels of experience and skill.

Demographic Control: A preliminary analysis was conducted to confirm that there were no significant differences in age and anthropometric characteristics between the control group (CG) and the experimental group (EG). This ensured that the groups were comparable and that any observed effects could be attributed to the experimental conditions rather than baseline differences.

                               Table 1. Anthropometric characteristics.

Parametrers

Experimental Group (n=12)

Control

Group (n=12)

p-value

Age

20.3 ± 1.1

20.4 ±1

0.686

Height (cm)

175.2 ± 5.1

176.1 ± 4.9

0.657

Body mass (kg)

70.0 ± 7.3

68.7 ± 5.8

0.647

                                Values are expressed as mean ± standard deviation.

  1. Sample Size Calculation: The sample size of 24 players (12 per group) was determined based on the minimum number needed to achieve adequate statistical power and reliable results. Previous studies and literature reviews guided the preliminary estimation of the required sample size to detect significant differences between groups, aiming to minimize the risk of Type I and Type II errors.
  2. Exclusion Criteria: Participants were excluded if they had recent injuries requiring medical attention, compromised health, neurological adverse events (e.g., seizures), incomplete participation in the training and testing program, or had a COVID-19 infection in the six months prior to the study. These criteria helped maintain sample homogeneity and representativeness, ensuring participants were in optimal condition for analysis.
  3. Statistical Analysis: Statistical tests were used to evaluate the representativeness and comparability of the groups by analyzing demographic and anthropometric characteristics, ensuring that the groups did not differ significantly on key variables. The results of statistical tests (t-test) showed that there were no significant differences between the groups in the baseline variables (p > 0.05), indicating that the groups were comparable at the beginning of the study." This confirmed that observed differences in results could be attributed to the experimental interventions rather than confounding factors.

These measures ensured that the sample was representative and that the study results could be generalized to the population of female volleyball players with similar characteristics.

Reviewer 2

  1. Results

- the tables and figures must be redone, for clarity and to respect JFMK MDPI templates

Reply

Dear reviewer, thank you for your suggestion. We have modified the Table as follows:

                               Table 1. Anthropometric characteristics.

Parametrers

Experimental Group (n=12)

Control

Group (n=12)

p-value

Age

20.3 ± 1.1

20.4 ±1

0.686

Height (cm)

175.2 ± 5.1

176.1 ± 4.9

0.657

Body mass (kg)

70.0 ± 7.3

68.7 ± 5.8

0.647

Values are expressed as mean ± standard deviation.

                      Table 2. Overview of the overall weekly training schedule.

Monday

Tuesday

Wednesday

Thursday

Friday

Saturday

Morning session

from 9.30

to  11.00 a.m.

Physical training

Motor reaction, agility and quickness.

Physical training

Strength training

Physical training

Motor reaction, agility and quickness.

Physical training

Strength training

Physical training

Motor reaction, agility and quickness.

Strength training

and friendly match on the sand

Afternoon

session

from 7.00 to 9.00

p.m.

Technical and  tactical training

Rest

Technical and  tactical training

Rest

Technical and  tactical training

Rest

Table 3. T-test results for independent groups EG and CG in RTsUL,RTsLL, TUL, TLL in pre and in post intervention (n=24).

Test

Group

X ±SD

DX

95% CI Lower

Upper

t

p

d

RTsUL (s)

pre

EG (n=12)

.482 ±.008

-.002

-.011

.006

-.59

.562

.18

CG  (n=12)

.484 ±.011

RTsUL (s)

post

EG (n=12)

.410 ±.019

-.062

-.077

-.049

-9.4

.000*

4.77

CG  (n=12)

.472 ±.013

RTsLL(s)

pre

EG (n=12)

.611 ±.027

-.007

-.029

.017

-.56

.582

.26

CG  (n=12)

.618 ±.027

RTsLL (s)

post

EG (n=12)

.520 ±.032

-.07

-.096

-.043

-5.53

.000*

2.41

CG  (n=12)

.590 ±.029

TUL (s)

pre

EG (n=12)

5.860 ±.244

.059

-.158

.275

.56

.580

.22

CG  (n=12)

5.801 ±.266

TUL (s)

post

EG (n=12)

5.238 ±.232

-.432

-.633

-.231

-4.46

.000*

1.78

CG  (n=12)

5.670 ±.243

TLL (s)

pre

EG (n=12)

9.153 ±.489

.05

-.345

.445

.26

.795

.11

CG  (n=12)

9.103 ±.444

TLL (s)

post

EG (n=12)

8.170 ±.473

-.840

-1.202

-.478

-4.81

.000*

2.23

CG  (n=12)

9.010 ±.376

RTsUL—Reaction Time simple Upper Limb; RTsLL—Reaction Time simple Lower Limb; TUL—Tapping Upper Limb;  TLL—Tapping Lower Limb; EG-experiment group; CG- control group; X ± DS—mean ± standard deviation; DX—difference of means; 95% C.I., interval of confidence with lower and upper levels;  t—Student’s t-test; p—statistical level of probability; d—Cohen ‘s effect size. * Significant at p<.05; T—value of t at the significance level of .05 = 2.201.

Table 4. T-test results for paired samples EG and CG in RTsUL,RTsLL, TUL, TLL in pre - post intervention (n=12).

Test

Group

Phase of Test

X

±SD

DX

post-pre

±SD

95% CI

Lower Upper

t

p

d

ip%

RTsUL (s)

EG

(n=12)

pre

.482 ±.008

.072

±.011

.065

.079

23.2

.000*

6.7

-14.9

post

.410 ±.019

CG

(n=12)

pre

.484 ±.011

-.012

±.004

.009

.015

9.6

.000*

2.8

-2.5

post

.472 ±.013

RTsLL (s)

EG

(n=12)

pre

.611 ±.027

-.091

±.007

.086

.096

44.0

.000*

12.7

-14.9

post

.520 ±.032

CG

(n=12)

pre

.618 ±.027

-.028

±.009

.022

.033

11.0

.000*

3.2

-4.5

post

.590 ±.029

EG

(n=12)

pre

5.860 ±.244

-.622

±.051

.589

.654

42.0

.000*

12.1

-10.6

TUL

 (s)

post

5.238 ±.232

CG

(n=12)

pre

5.801 ±.266

-.131

±.073

.085

.177

6.2

.000*

1.8

-2.3

post

5.670 ±.243

EG

(n=12)

pre

9.153 ±.489

-.983

±.079

.932

1.033

43.1

.000*

12.4

-10.7

TLL

 (s)

post

8.170 ±.473

CG

(n=12)

pre

9.103 ±.444

-.093

±.115

.019

.166

2.8

.018*

.8

-1.0

post

9.010 ±.376

RTsUL—Reaction Time simplevUpper Limb; RTsLL—Reaction Time simple Lower Limb; TUL—Tapping Upper Limb;  TLL—Tapping Lower Limb; EG-experiment group; CG- control group; X ± DS—mean ± standard deviation; DX—difference of means; 95% C.I., interval of confidence with lower and upper levels;  t—Student’s t-test; p—statistical level of probability; d—Cohen ‘s effect size; ip%—  increase percentage; * Significant at p<0.05;   T—value of t at the significance level of 0.05 = 2.074.

Reviewer 2

. Discussion

- L287: the authors again use the expression "Numerous studies,..." and they argue with a single citation, of a single study. I request the removal of the expression "numerous studies" if the authors did not refer to several studies.

Reply

Dear reviewer, we have corrected as follows:

Hodges et al., suggest that the speed of movement does not only depend on the physical abilities of individuals, but also on cognitive factors that play a crucial role [23].

Reviewer 2

L295: clarify why you use "physical activity (PA) sensors" and "perception technological tools (PAD)" - do the authors differentiate the two means or do they refer to the same category of instruments, but named differently???

Reply

Dear reviewer, we have corrected as follows:

From a methodological standpoint, our results confirm that a 6-week training program incorporating PAD sensors is sufficient to enhance specific Performance-Related Cognitive Functions (PCFs).

Reviewer 2

- I think that the first conclusion should be redone, adapted, differently formulated because the variable targeted by the study was the introduction of exercises with sensor technology (PAD) and not the 6-week training!!

Reply

Dear reviewer, we have corrected as follows:

The use of light-based perception-action technology devices, integrated into a 6-week training program, led to significant improvements in both movement quickness and motor reaction speed among female volleyball players. These advanced technological tools provided immediate, precise feedback and stimulated athletes' visual and cognitive systems, enabling them to react more swiftly and efficiently to external stimuli. This enhanced training regimen proved to be more effective than traditional methods, demonstrating not only its potential to accelerate skill acquisition but also its ability to fine-tune the athletes' neuromuscular coordination. By refining these critical aspects of performance, the athletes will be able to execute volleyball techniques with greater speed and precision, gaining a competitive advantage on the court.

Reviewer 2

L351-360 : these are not conclusions, but rather practical implications of the study. I recommend a "practical implications" section

Reply

Dear reviewer, we have corrected as follows:

Practical implications and further research

Exercises enriched with PAD, when adapted to the specific performance model of the sport and employing appropriate methodologies, can produce superior results in certain contexts compared to conventional training approaches. This aspect is particularly pertinent for those involved in physical preparation, where there is often a critical need to rapidly enhance performance levels during the pre-season phase. Therefore, coaches and quickness and conditioning specialists should consider in-tegrating light perception-action sensors into their training methodologies to optimize sports performance. Nonetheless, further research is warranted, particularly to validate the long-term effects with larger athlete samples encompassing various sports disciplines and both genders. Additionally, controlled environment testing, such as laboratory-based assessments of physiological parameters, should be incorporated to aid in developing standardized testing protocols that incorporate perception-action technologies.

Reviewer 2

L361-364 repeats what was said in L345-348

- I recommend the reformulation of the conclusions, with greater highlighting of the essence of the title. If not, I reformulate the title.

- The conclusions, the purpose and the title do not converge. Please clarify this point

Reply

Dear Reviewer, we have modified the conclusions and at the same time the title as you suggested. Thank you.

Title: The impact of perception-action training devices on quickness and reaction time in female volleyball players

Reviewer 2

  1. References

- 35% of references are older than 12 years. I recommend the authors to bring up newer references on the topic

Reply

Dear Reviewer, we have modified. Thank you.

  1. Oliinyk, I.; Doroshenko, E.; Melnyk, M.; Tyshchenko, V.; Shamardin, V. Modern Approaches to Analysis of Technical and Tactical Actions of Skilled Volleyball Players. Teorìâ ta Metod. Fìzičnogo Vihovannâ 2021, 21, 235–243. DOI: 10.17309/tmfv.2021.3.07

  1. Salles, W. das N.; Collet, C.; Porath, M.; Milistetd, M.; Nascimento, J.V. do Factors Associated to Performance Efficacy of Technical-Tactical Actions in Volleyball. Rev. Bras. Cineantropometria Desempenho Hum. 2017, 19, 74–83. https://doi.org/10.5007/1980-0037.2017v19n1p74
  2. Imas,Y., Borysova,O., Dutchak, M., Shlonska,O., Kogut, I., Marynych, V. (2018). Technical and Tactical Preparation of Elite Athletes in Team Sports (Volleyball). DOI:10.7752/jpes.2018.02144
  3. Walankar, P., & Shetty, J. (2020). Speed agility and quickness training: A review. Int J Phys Educ Sports Health, 7(6), 157–159.
  4. Hodges NJ, Wyder-Hodge PA, Hetherington S, Baker J, Besler Z, Spering M. Topical Review: Perceptual-cognitive Skills, Methods, and Skill-based Comparisons in Interceptive Sports. Optom Vis Sci. 2021 Jul 1;98(7):681-695. doi: 10.1097/OPX.0000000000001727. PMID: 34328450.
  5. Wang J, Qin Z, Wei Z. Power and velocity performance of swing movement in the adolescent male volleyball players - age and positional difference. BMC Sports Sci Med Rehabil. 2024 May 16;16(1):111. doi: 10.1186/s13102-024-00898-2. PMID: 38755687; PMCID: PMC11097490.
  6. Mawarti, S., Rohmansyah, N. A., & Hiruntrakul, A. (2021). Effect of volleyball training program to improve reaction time. Int J Human Movement Sports Sci, 9(6), 1314-8. DOI: 10.13189/saj.2021.090627
  7. Badau, D., Badau, A., Ene-Voiculescu, C., Larion, A., Ene-Voiculescu, V., Mihaila, I., ... & Abramiuc, A. (2022). The impact of implementing an exergame program on the level of reaction time optimization in Handball, volleyball, and basketball players. International Journal of Environmental Research and Public Health, 19(9), 5598. https://doi.org/10.3390/ijerph19095598

  1. Badau D, Badau A. Optimizing Reaction Time in Relation to Manual and Foot Laterality in Children Using the Fitlight Technological Systems. Sensors (Basel). 2022 Nov 14;22(22):8785. doi: 10.3390/s22228785. PMID: 36433379; PMCID: PMC9694787.

  1. Theofilou G, Ladakis I, Mavroidi C, Kilintzis V, Mirachtsis T, Chouvarda I, Kouidi E. The Effects of a Visual Stimuli Training Program on Reaction Time, Cognitive Function, and Fitness in Young Soccer Players. Sensors (Basel). 2022 Sep 3;22(17):6680. doi: 10.3390/s22176680. PMID: 36081136; PMCID: PMC9460176.

Reviewer 2

  1. Technical aspects

- Reference no. 10 has the year "2020" crossed out 2 times and the names of the authors, the title are doubled - I recommend correcting and respecting the MDPI format

- The clarity of figures 2 and 3 must be increased

- The tables do not respect the format, being more inserted figures than constructed tables.

- Few references, considering the wealth of sources on this subject

Reply

Dear Reviewer, thank you for your comment. We have modified following your suggestion

Round 2

Reviewer 1 Report

Comments and Suggestions for Authors

It is clear from this version that the authors tried to improve and meet the suggestions, but not everything that was requested was met. Furthermore, the commentary boxes in the text are in Chinese and very hard to read.

The authors continue to fail to indicate how the sample size was calculated. My suggestion regarding the title was not met.

The abstract has improved a lot, but the text needs to be proofread by an English editor first, it’s possible to perceived typo errors such as “sommary”.

The introduction still lacks a clear aim and there is no hypothesis as previously suggested.

There is one point where the authors should be a little more cautious. It is stated at the end of the abstract: “, the use of light-based perception-action technology devices, integrated into a 6-week training program, led to significant improvements in both movement quickness and motor reaction speed among female volleyball players, demonstrating greater effectiveness compared to traditional training methods.” One should be careful before making this type of statement. Will the improvement in reaction time and agility really lead to an increase in volleyball performance? One should be cautious about the scope of the study's results. As the authors themselves state in the introduction: “Players require moderate to high levels of sensory and cognitive skills, in addition to physical and motor abilities, as fundamental prerequisites” If we consider that the participants play in the Italian league, which is one of the strongest leagues in the world, will this type of training really add anything? It is possible that players at this level at which they were measured in this study already have moderate levels of sensory and cognitive skills.

To improve the quality of results, I suggest that the authors present the reliability indicators of the tests applied. This can be reported in the studies that were cited as references.

A curiosity: why did the authors use two tests for the lower limbs and one for the upper limb?

All Tables and Figures must be self-explanatory. If the authors use an abbreviation such as “TUL”, it must be explained in the caption.

Comments on the Quality of English Language

Need improve.

Author Response

Reviewer 2

It is clear from this version that the authors tried to improve and meet the suggestions, but not everything that was requested was met. Furthermore, the commentary boxes in the text are in Chinese and very hard to read.

Reply

Dear Reviewer, thank you for your careful and thorough response. We regret that we were unable to improve and meet all your suggestions, but we are doing our best to achieve a successful outcome. Below you will find point by point answers to all your suggestions.

Reviewer 2

The authors continue to fail to indicate how the sample size was calculated.

Reply

Dear reviewer, thank you for your comment. To select the study cohort, we performed a bibliographical search to evaluate the number of similar studies, which provided us with an indicative optimal sample of 20/25 subjects. We then performed a test to evaluate the statistical power relative to the number of subjects we had recruited, in our case 24. The result of the test provided us with an effect size of 0.802 and therefore we continued with the study. We also reported this data in the materials and methods section.

Reviewer 2

My suggestion regarding the title was not met.

Reply

Dear Reviewer, as you suggested, the title has already been modified to make it clearer and more specific to the study’s objective.
First title: Effects of action perception light sensors on volleyball players performances
Modified title: The impact of perception-action training devices on quickness and reaction time in female volleyball players.

Reviewer 2

The abstract has improved a lot, but the text needs to be proofread by an English editor first, it’s possible to perceived typo errors such as “sommary”.

Reply

Dear Reviewer, we apologize for the inconvenience. We have revised the grammar in the abstract and throughout the manuscript.

Reviewer 2

The introduction still lacks a clear aim and there is no hypothesis as previously suggested.

Reply

Dear Reviewer, thank you for your comment. Below are the changes made:

  1. Introduction

This study aims to explore how employing advanced technologies, such as action perception devices (PAD), can enhance training effectiveness and positively impact athletic performance in volleyball. The main hypothesis is that a training program enriched with specific exercises using technological tools can significantly improve reaction times and quickness of movement compared to a traditional training program for elite volleyball players. Specifically, it is expected that such technological tools can provide immediate feedback and visual stimuli that facilitate a faster and measurable improvement in reaction capabilities and quickness of movement.

The objective of this study was to examine the performance differences between a training program that utilizes PAD (Perception-Action Technology) tools and a traditional training program (without the use of advanced technologies) on motor reaction times and quickness of the upper and lower limbs in volleyball players. The main hypothesis is that a training program enriched with specific exercises using technological tools can significantly improve reaction times and quickness of movement compared to a traditional training program for elite volleyball players. Specifically, it is expected that such technological tools can provide immediate feedback and visual stimuli that facilitate a faster and measurable improvement in reaction capabilities and quickness of movement.

Reviewer 2

There is one point where the authors should be a little more cautious. It is stated at the end of the abstract: “, the use of light-based perception-action technology devices, integrated into a 6-week training program, led to significant improvements in both movement quickness and motor reaction speed among female volleyball players, demonstrating greater effectiveness compared to traditional training methods.” One should be careful before making this type of statement. Will the improvement in reaction time and agility really lead to an increase in volleyball performance? One should be cautious about the scope of the study's results. 

Reply

Dear Reviewer,

Thank you for your comment and for your attention to the details of our research. We appreciate your suggestion and agree that while our results indicate significant improvements in movement speed and reaction time within the scope of our tests, it is essential to be cautious about the broader implications of these improvements for overall performance in competitive contexts. Our statement regarding the greater effectiveness of technological devices compared to traditional methods may indeed be premature without further direct evidence demonstrating a concrete impact on performance in matches. Considering this, we will make the following revisions:

Abstract

In summary, the use of light-based perception-action technology devices, integrated into a 6-week training program, led to significant improvements in both movement quickness and motor reaction speed among female volleyball players, demonstrating greater effectiveness compared to traditional training methods.

The use of light-based perception-action technology devices in volleyball training has shown potential for significantly improving movement speed and reaction time. However, further research is needed to determine whether these improvements actually translate into enhanced overall performance in competitive contexts compared to traditional training methods.

  1. Conclusions

The use of light-based perception-action technology devices, integrated into a 6-week training program, led to significant improvements in both movement quickness and motor reaction speed among female volleyball players. These advanced technological tools provided immediate, precise feedback and stimulated athletes' visual and cognitive systems, enabling them to react more swiftly and efficiently to external stimuli.This enhanced training regimen proved to be more effective than traditional methods, demonstrating not only its potential to accelerate skill acquisition but also its ability to fine-tune the athletes' neuromuscular coordination. By refining these critical aspects of performance, the athletes will be able to execute volleyball techniques with greater speed and precision, gaining a competitive advantage on the court.

The use of light-based perception-action technology devices, integrated into a 6-week training program, has led to significant improvements in both movement speed and motor reaction time tests among volleyball players. These advanced technological tools provided immediate and precise feedback, stimulating the athletes' visual and cognitive systems, allowing them to react more quickly and efficiently to external stimuli. By refining these critical performance aspects, it is expected that athletes will be able to execute volleyball techniques with greater speed and accuracy, gaining a competitive advantage on the field. However, further research is needed to determine whether these improvements translate into overall enhanced performance in competitive contexts compared to traditional training methods.

 Reviewer 2

As the authors themselves state in the introduction: “Players require moderate to high levels of sensory and cognitive skills, in addition to physical and motor abilities, as fundamental prerequisites” If we consider that the participants play in the Italian league, which is one of the strongest leagues in the world, will this type of training really add anything? It is possible that players at this level at which they were measured in this study already have moderate levels of sensory and cognitive skills.

Reply

Dear Reviewer,

Thank you for your pertinent observation regarding the skill level of the participants and the effectiveness of the type of training tested. We agree that players participating in top-level leagues, such as the Italian championship, already possess high levels of sensory and cognitive skills, in addition to their physical and motor abilities. Although these players have already reached a high level of skill, our study seeks to determine if advanced technologies can provide additional marginal benefits and contribute to our understanding of how to optimize training for elite athletes. However, in this study we are dealing with the part relating to athletic preparation, that is, a work to improve the physical efficiency of functional athletes to have a subsequent transfer in the specific technique of volleyball, as specified in 2.2. Experimental procedure and in Table 2 Overview of the overall weekly training schedule.

Thank you again for your valuable feedback.

 Reviewer 2

To improve the quality of results, I suggest that the authors present the reliability indicators of the tests applied. This can be reported in the studies that were cited as references.

Reply

Dear Reviewer, thank you for your pertinent observation.

while for the motor reaction tests the reliability indices were already reported:

The simple upper limb reaction time test (RTsUL; ICC/Rho: 0.94, 95% CI: 0.72–0.98, p < .001) [29]

The Reaction Time simple lower limb test (RTsLL; ICC/Rho: 0.74 * (95% CI: 0.34 and 0.92), p < .001) [30]

For the TUL and TLL tests we modified as follows:

2.6. Tapping Lower Limb (TLL)

The test is designed to assess how quickly the feet can move [33].

The test is designed to assess how quickly the feet can move, (ICC > 0,90) [32]

References

[32] Chaabouni, S., Methnani, R., Al Hadabi, B., Al Busafi, M., Al Kitani, M., Al Jadidi, K., ... & Gmada, N. (2022). A simple field tapping test for evaluating frequency qualities of the lower limb neuromuscular system in soccer players: A validity and reliability study. International journal of environmental research and public health, 19(7), 3792.

2.7. Tapping Upper Limb (TUL)

The test aims to measure the quickness and coordination of movement of the upper limb [32]

The test aims to measure the quickness and coordination of movement of the upper limb [35],  (ICC 0.88-0.93)

[35] Di Libero T, et al. An Overall Automated Architecture Based on the Tapping Test Measurement Protocol: Hand Dexterity Assessment through an Innovative Objective Method., Sensors (Basel). 2024. PMID: 39000912

Reviewer 2

A curiosity: why did the authors use two tests for the lower limbs and one for the upper limb?

Reply

Dear Reviewer, thank you for your observation.

We found that we used 4 tests, 2 motor reaction tests, 1 for the upper limbs (RTsUL - Reaction Time simple Upper Limb) and one for the lower limbs (RTsLL - Reaction Time simple Lower Limb) and 2 quickness tests, 1 for the upper limbs (TUL - Tapping Upper Limb) and one for the lower limbs (TLL - Tapping Lower Limb)

Reviewer 2

All Tables and Figures must be self-explanatory. If the authors use an abbreviation such as “TUL”, it must be explained in the caption.

Reply

Dear Reviewer, thank you for your observation. We have checked the captions of all tables and figures and made the changes you suggested.

Dear reviewer, furthermore, to make the manuscript with the changes more understandable, we have carried out a total check and simplified the insertion of the new parts and corrected the grammar.

Thank you again for your valuable feedback.

Round 3

Reviewer 1 Report

Comments and Suggestions for Authors

In the current version, the authors have adequately adjusted the manuscript and it is ready to be published by JFMK.